# A Study of Compressibility, Compactability and Mucoadhesivity of Tableting Materials for Matrix Systems Based on Chitosan

**DOI:** 10.3390/polym13213636

**Published:** 2021-10-21

**Authors:** Jitka Muzikova, Eva Snejdrova, Juraj Martiska, Bara Doubkova, Andrea Veris

**Affiliations:** 1Department of Pharmaceutical Technology, Faculty of Pharmacy, Charles University, Akademika Heyrovskeho 1203, 50005 Hradec Kralove, Czech Republic; muzikova@faf.cuni.cz; 2InStar Technologies a.s., Mrstikova 399/2a, 46007 Liberec, Czech Republic; martiska@instar.tech; 3Dr. Müller Pharma s.r.o., U Mostku 182, 50341 Hradec Kralove, Czech Republic; doubkova@muller-pharma.cz (B.D.); veris@muller-pharma.cz (A.V.)

**Keywords:** chitosan, matrix tablets, silicified microcrystalline cellulose, compressibility, compactability, mucoadhesion

## Abstract

The objective of the present research is to evaluate directly compressible chitosan-based tableting materials for the formulation of mucoadhesive matrix tablets intended for targeted drug release to distal segments of the GIT. The influence of sodium alginate, hypromellose, and silicified microcrystalline cellulose (P90) on compressibility, compactability and lubricant sensitivity ratio was tested. Furthermore, the rheological properties of the hydrated surface layer of the matrix tablets and the mucoadhesion to a mucin substrate were analysed. Compressibility was evaluated using the energy profile of the compression process, compactability by means of the tensile strength of tablets, and lubricant sensitivity ratio was calculated to assess the sensitivity to lubricant. Addition of P90 to chitosan improved compressibility, which is demonstrated by the increase in the energy of plastic deformation and the higher tensile strength of tablets. P90 also significantly reduced the high lubricant sensitivity of chitosan. Presence of retarding components led to a decrease in Emax. All tested matrix tablets revealed a good mucoadhesion without a negative effect of P90 content. The viscosity of a gel layer on the surface of matrix tablets containing hypromellose was higher compared to those with sodium alginate. This was not reflected in the adhesive strength of the tablets. The formulated tableting materials combining chitosan and P90 are a suitable matrix for incorporation of an active ingredient, whose delayed release in the intestine can be achieved by the functionality of the chitosan-sodium alginate complex.

## 1. Introduction

Natural polysaccharides and their analogues are substances with a great potential for use in the pharmaceutical industry. In solid dosage forms they have been employed as excipients in the delivery systems with a modified release. With these oral systems, a targeted release of the active ingredient in the distal segment of the gastrointestinal tract can be achieved [1,2,3,4,5,6]. One of these polysaccharides is chitosan, whose safety, nontoxicity, biocompatibility, and biodegradability predetermine it for biomedical applications [7]. Over the years, several fields of application were described, including tissue engineering, genotherapy, manufacturing antibacterial and hemostatic materials, and, last but not least, the formulation of targeted drug delivery systems, especially thin films, particulate forms, and tablets [8,9,10].

Chitosan, chemically (1→4)—2 amino-2-deoxy-β-D-glucan, ranks among cation-active polysaccharides and it is a linear copolymer composed of acetylated and deacetylated units connected with β-(1→4) glycosidic bonds [10,11,12,13]. It is obtained by deacetylation of chitin, its abundant natural resource. The degree of deacetylation is usually 60–100% and fundamentally affects its properties. Owing to ionization, chitosan is soluble in water at pH < 6 and above this value it becomes water insoluble. Therefore, chitosan can be used as a viscosifier in acidic media with the viscosity of its solutions increasing with the degree of deacetylation [14]. The presence of hydroxyl and amino functional groups allows interactions with proteins, macromolecular and low molecular weight active substances, which leads to a higher loading capacity of the therapeutic system. As a result of the presence of a reactive positively charged amino group, it exerts antimicrobial activity against most gram-negative and gram-positive bacteria and fungi at a pH < 6 [15]. Chitosan has a strong ability to adhere to mucous tissues, which is widely used in the formulation of a variety of adhesive delivery systems for controlled drug release. In these systems, the targeting excipients ensure delivery of the drug into the desired site and the adhesion ensures its sustained release. This mechanism of action has been successfully used to achieve the targeted release of the active ingredient in the distal segment of the GIT [16,17].

Cation-active chitosan can be combined with an anion-active polysaccharide to produce a polyelectrolyte complex used in matrix tablets with controlled release of the active ingredient. An example of an anion-active polysaccharide is sodium alginate, which is also obtained from a natural source, sea algae. Unlike chitosan it is water-soluble in alkaline media and becomes insoluble in milieu with pH < 3. A matrix tablet containing a combination of these acts after oral administration as a film-coated hydrophilic gel system. Drug release occurs on the basis of erosion and swelling of the tablets. In acidic pH, alginate is unionized and insoluble. However, ionization and solubility of chitosan causes polymer degradation and water absorption into the tablets. At a higher pH, the polymers begin to interact with each other and form a polyelectrolyte complex film on the surface of tablets, which decreases erosion and decelerates swelling. Thus, the mechanism of drug release is different than in the traditional matrix tablets [13,18,19,20].

Chitosan exhibits poor flowability and compressibility, which limits its use in tableting materials for direct compressing [21]. Its consolidation and compression behaviour shows lower plasticity and higher elasticity than in microcrystalline cellulose [22]. Moreover, the degree of chitosan deacetylation influences the tablet strength [21]. Chitosan with a low degree of deacetylation produces tablets with better mechanical properties than chitosan with a high degree of deacetylation, which is connected with a lower degree of polymerization [23]. For direct compression, it is advantageous to combine chitosan with another dry binder, e.g., Avicel PH 200 or Prosolv SMCC 90 (P90). The proven ratio of chitosan to the substances is 70:30 [24].

This study was designed to assess the compressibility, compactability, lubricant sensitivity. and mucoadhesion of directly compressible chitosan-based tableting materials for the formulation of tablets providing targeted drug release to distal segment of the gastrointestinal tract. The influence of silicified microcrystalline cellulose Prosolv SMCC 90 used as a dry binder and sodium alginate and hypromellose acting as retarding components was studied. Compressibility was evaluated using the energy profile of the compression process, compactability by means of the tensile strength of tablets, and lubricant sensitivity by LSR values. The rheological properties of the hydrated surface layer of the matrix tablets and the mucoadhesion to mucin substrate were tested by rotational and tensile tests on an absolute rheometer.

## 2. Materials and Methods

### 2.1. Materials

The study used the substances chitosan (JBICHEM, Zhoushan, China), sodium alginate (Merck, Darmstadt, Germany), hypromellose Methocel K100M (Colorcon GmbH, Idstein, Germany), and silicificated microcrystalline cellulose Prosolv SMCC 90 (JRS PHARMA Gmbh + Co.KG, Rosenberg, Germany). The lubricant being magnesium stearate (ACROS Organics, Branchburg, NJ, USA). Phosphate buffered saline pH 6.8 (Ph.Eur.) was used as medium for preparation of tableting mixture hydrogels and model mucous substrate and for wetting of the matrix tablets. Mucin from porcine stomach Type II was purchased from Merck, Darmstadt, Germany, and used as model substrate for adhesion testing.

### 2.2. Preparation of Tableting Materials

The study investigated 28 tableting materials, whose composition is shown in Table 1 and Table 2. A mixing cube KB 15S (Erweka GmbH, Langen, Germany) was used for the mixture preparations. Tableting materials were prepared by graded mixing. A mixture of chitosan and P90 in the ratio of 3:1 was prepared by mixing the substances for a period of 2 min. Mixtures of chitosan (or chitosan + P90) and alginate with or without hypromellose were prepared by mixing for 3 min. Finally, magnesium stearate was added with a period of mixing of 2 min. 

### 2.3. Preparation of Tablets and Energy Evaluation of Compression Process

Tablets were compressed using a T1 FRO 50 TH.A1K Zwick/Roell device (Zwick GmbH & Co. KG, Ulm, Germany) equipped with a special die with a lower and an upper punch. The rate of compression was 40 mm/min, the preload 2 N, and the rate of preload 2 mm/s. The tablets were cylindrical without facets and had a diameter of 13 mm and a weight of 0.100 ± 0.001 g. From each tableting material 10 tablets were compressed at the compression force of 4 kN, which provided sufficient tablet strength. Energy profiles of compression during tablet preparation were calculated by press controlling program testXpert V 9.01 (Zwick GmbH & Co. KG, Ulm, Germany). Evaluated parameters were E1—precompression energy (J), E2—energy of plastic deformation (J), E3—energy of elastic deformation (J), Emax—total energy (J), and Pl—plasticity (%) [25,26]

### 2.4. Evaluation of Compactability and Lubricant Sensitivity

Compactability was evaluated by determining the tensile strength of the tablets. Thickness and diameter of 10 tablets were measured with a precision of 0.01 mm using the Tablet Tester M8 (Dr. Schleuniger Pharmatron AG, Switzerland). Subsequently, the destruction force in N was measured with the same device. Tablets were tested no sooner than 24 h after compression. The following equation (Equation (1)) was used for calculation of tensile strength according to Fell and Newton [27]:(1)$$P=\frac{2F}{\pi \mathrm{dh}}$$
where P (MPa) is tensile strength of tablets, F (N) is destruction force, d (mm) is the diameter of tablets, and h (mm) is the height of tablets.

The mean values of tensile strengths were used for calculation of lubricant sensitivity ratio (LSR), from which dry binder can be compared in terms of sensitivity to added lubricants. It can be calculated according to the equation (Equation (2)) [28]:(2)$$\mathrm{LSR}=\frac{\left(\mathrm{Csu}-\mathrm{Csl}\right)}{\mathrm{Csu}}$$
where Csu is strength of tablets without a lubricant and Csl is strength of tablets with a lubricant.

In this paper, the values of tensile strength instead of crushing strength are used for LSR calculation. The tensile strength is used to increase the precision of evaluation, as tablet dimensions are included in its calculation.

### 2.5. Rheological Testing

Rheological properties of the dispersions in phosphate buffered saline (PBS) pH 6.8 of retarding excipients alone and selected tableting mixtures perspective in terms of controlled drug release were evaluated by performing an equilibrium shear rate test. The resultant viscosity curves were analysed by fitting to the power law model. A Kinexus rotational rheometer with a Peltier plate cartridge and CP 2/20 measuring system, and standard pre-configured sequence Viscometry_0010 Table of shear rates with the power law model fit in the rSpace for Kinexus software version 1.76 were used. The measurements were performed at 37 °C and shear rate range was between 0.1 and 100 s^−1^. Flow behaviour of the samples was evaluated by fitting of the viscosity curves by a power law model (Equation (3)).
(3)$$\eta ={K*D}^{n-1}$$
where η (Pa s) is shear viscosity, D (s^−1^) is shear rate, K (Pa s^n^) is consistency index, and n (−) is power law index.

The consistency index K (Pa s^n^) numerically equals to the viscosity measured at shear rate 1 s^−1^; power law index n ranges from 0 for highly shear thinning materials to 1 for Newtonian materials. All the measurements were done in triplicate, and averages and standard deviations were calculated.

### 2.6. Mucoadhesion Testing

Adhesive properties of matrix tablets of selected compositions after exposing to PBS pH 6.8 were subjected to tensile test using a Kinexus rotational rheometer with a Peltier plate cartridge using matched PU 20 mm and modified sequence rSolution_0020 evaluating tackiness and adhesion using a pull away test in the rSpace for Kinexus software version 1.76. A standard loading sequence was used to ensure samples were subjected to a consistent and controllable loading protocol, with a working gap of 1 mm employed and sample trimmed flush with the plate edge. The tests were performed at 37 °C using a mucin from porcine stomach as a model substrate. Powdered mucin was hydrated by sufficient amount of PBS pH 6.8 to reach a standard model substrate with viscoelastic properties suitable for adhesion testing. The adhesive properties of the samples were evaluated as the peak in normal force F_max_ (N). All measurements were done five times, the averages and standard deviations were calculated.

### 2.7. Statistical Analysis

For the statistical evaluation of the results the program MS Excel was used. In the case of similar significance of values, an ANOVA test at the level of significance of 0.05 was employed. In the following text, “statistically insignificant” results are those for which the *p*-value is higher than 0.05.

## 3. Results and Discussion

### 3.1. Evaluation of Compressibility

Compressibility of chitosan-based tableting materials was evaluated by means of the energy profile of the compression process. The values of individual energies are presented in Table 3 and Table 4. 

Emax is the total energy of compression equalling to the sum of precompression energy (E1), plastic deformation energy (E2), and elastic deformation energy (E3). The values of these energies describe the compression process from an energy point of view and allow us to compare the compressibility of different tableting materials. Energy of precompression is important for the particle rearrangement during the precompression phase and it is associated with different properties of particles (size, shape, mechanism of compression). Lower values of this energy are preferred. Energy of plastic deformation is important for the bonding and for the strength of tablets. This energy is accumulated in tablets after the compression. The higher the energy of plastic deformation, the higher the strength of tablets. Energy of elastic deformation is released during the decompression phase. It is advantageous if the values of this energy are low [25,26].

The highest values of Emax were achieved with chitosan and its mixture with P90. An addition of a retarding component, i.e., sodium alginate or its mixture with hypromellose 100M, led to a decrease of Emax. This drop was greater in the case of alginate alone and showed concentration dependency (lowest concentration, i.e., 30% corresponds to the lowest energy drop). The influence of the concentration of retarding components on Emax is statistically insignificant. An addition of the lubricant magnesium stearate decreased the values of total energy of compression with the exception of the tableting materials with hypromellose 100M. The values decreased with increasing concentration of the retarding component.

The values of the energy of precompression (E1) decreased with the addition of retarding components to chitosan, the energy decrease was more significant after the addition of sodium alginate due to its presence in higher concentration. Considering the tableting materials with sodium alginate and HPMC100M, no statistically significant difference in the E1 values was recorded within the contained concentration. Mixtures of chitosan with P90 showed higher values of this energy except in the tableting materials with a combination of retarding components. The lubricant magnesium stearate decreased the energy of precompression. Its influence was most significant in matrices with chitosan alone, followed by the tableting materials with sodium alginate. The impact of the lubricant was not seen in the mixtures with a combination of retarding components. 

The highest values of energy of plastic deformation (E2), which is of great importance for the formation of bonds, were observed in the mixtures with P90. Silicified microcrystalline cellulose exerts good compressibility, and the mechanism of compression is plastic deformation [29,30,31]. After the addition of the lubricant to the tableting materials, a decrease in the values of the E2 was observed. Energy of elastic deformation was slightly higher in the case of mixtures without P90 and decreased with the addition of retarding components to both chitosan and its mixture with P90. The addition of sodium alginate did not lead to the statistically significant changes in its values within the concentrations used. The same is valid for tableting materials with a mixture of retardants except those where a 30% mixture was added to chitosan with P90. The energy of elastic deformation was slightly increased by the presence of lubricant in the mixtures with P90 except in the mixtures without retarding components and the mixtures with a 30% combination of alginate and HPMC 100. 

There are no more marked differences between the values of plasticity; a decrease in plasticity was recorded only in the case of the tableting materials with a mixture of chitosan and P90 with sodium alginate.

### 3.2. Evaluation of Compactability and Lubricant Sensitivity

Compactability of tableting materials was evaluated using the tensile strength of tablets. The results of this evaluation are presented in Table 5.

The highest values of tensile strength of tables were achieved by chitosan in combination with P90 and its mixtures with a combination of sodium alginate and HPMC100M in the ratio of 1:1. Addition of sodium alginate markedly decreased the values of the tensile strength of tablets. The presence of lubricant caused a further decrease in tensile strength of tablets in all tableting materials, but most significantly in chitosan alone. The addition of P90 to chitosan reduced its lubricant sensitivity, as colloidal silicon dioxide competitively inhibits the binding sites for magnesium stearate [32,33]. If retarding components were present in the mixture, the amount of P90 was lower and with that also its effect on decreasing lubricant sensitivity. This is obvious from the calculated LSR values, which are shown in Table 5. LSR values range between 0–1. The more the LSR values approach 1, the more sensitive the tableting material is to the added lubricant [28]. Based on the presented values, the highest sensitivity was exerted by chitosan alone, which showed clearly plastic deformation [21]. Considering the tablets with chitosan, the addition of sodium alginate decreased lubricant sensitivity, which means that softening of tablets by the action of the lubricant is lower. The addition of sodium alginate in combination with HPMC in a 1:1 ratio led to a further reduction of this sensitivity with the exception of the tableting material with a 50% share of this mixture. The lowest LSR value was observed by the mixture of chitosan and P90 in the ratio of 3:1, where colloidal silicon dioxide intervenes into the mechanism of adhesion of lubricant to microcrystalline cellulose. An addition of sodium alginate into tableting material increased the sensitivity to the lubricant with its increasing concentration. In the case of the addition of HPMC100M, the lowest sensitivity was shown by the tableting material with 30% of retarding mixture. With increasing concentration of this mixture, the LSR values grew because of the increasing share of sodium alginate in tableting material.

### 3.3. Rheological and Mucoadhesion Testing

Drug release from matrix tablets and consequently the final effect of the medicinal product targeted to the intestine is influenced by the rheological and adhesive properties of the polymeric carriers after the treatment of physiological fluids in the GIT. The viscosity and mucoadhesion of tableting materials after exposure to PBS pH 6.8 were studied to identify the optimal compositions for targeting the drug to the intestine and to ensure a suitable drug release profile. The courses of the viscosity–shear rate curves clearly demonstrate typical shear-thinning behaviour as viscosity drops with increasing shear rate. The linear sections of the viscosity curves were fitted by the power law model, where the correlation coefficients are a good measure of how well the data fit the model. For all the samples the values were above 0.985, indicating good correlation between measured and predicted data. The power law coefficients were used to compare the rheological behaviour of the studied materials. The consistency index K, numerically equal to the viscosity at 1 s^−1^, serves as a good measure of a zero-shear viscosity (or at-rest-viscosity) for comparative purposes, and the index of non-Newtonian behaviour reflects the sensitivity of the material to a stress. 

The impact of 30%, 40%, or 50% of retarding components on viscosity were studied. As retarding components, either sodium alginate (SA) alone or its mixture with hypromellose in ratio 1:1 (SA/HPMC100M) were tested. In Figure 1, an increase in viscosity with increasing concentration of retardant is significant. However, substitution of 25% of chitosan by P90 in tableting mixtures caused a lower increase in viscosity. 

The mixture of SA with HMPC100M provide significantly more viscous systems compared to SA alone as HMPC100M is more effective gelling agent than SA, as shown in Figure 2.

The power law index n can be used for comparison of the viscoelastic properties of the tableting materials and prediction of the changes during the passage of the tablet in the intestine. Considering that n equals one for Newtonian materials, it can be established that all measured samples are fairly shear-thinning having the values of n in the narrow range of 0.4 to 0.6 (Table 6). It corresponds to the consistency K, meaning a more viscous or stiff gel layer is more sensitive to shearing [34]. 

Mucoadhesive properties of the chiton-based tablets were evaluated based on measuring the maximum force needed for detachment of the tablet hydrated with PBS pH 6.8 from the mucin substrate. As shown in Figure 3, all tested tablets revealed the mucoadhesive properties under the used test conditions.

This finding is consistent with the rheological and texture profile measurements, indicating a very good adhesion [35]. No significant difference between chitosan alone and mixture chitosan/P90 was detected. The concentration of the retarding component has a higher impact on mucoadhesion than the type of retardant used. However, in the case of concentrations of retardant at 30% and 40%, the combination of sodium alginate and hypromellose (SA/HPMC100M) ensures better mucoadhesion than sodium alginate alone. 

Significantly higher viscosity of the compositions with hypromellose was not reflected in the adhesive strength. An explanation may be the different mechanism of mucoadhesion of non-ionic hypromellose and anionic alginate [36]. Both these linear polysaccharides adhere to the intestinal mucosa thanks to the spreading of a sticky hydrogel formed after exposure to a physiological medium. However, mucoadhesion of anionic alginate is supported by carboxyl end groups favouring the formation of hydrogen bonds with mucin substrate. The polyanion polymers are considered more effective bioadhesives than nonionic polymers [37] and an increase in charge density can provide even better adhesion.

## 4. Conclusions

The directly compressible chitosan-based tableting materials for the formulation of mucoadhesive matrix tablets intended for targeted drug release to the distal segments of the GIT were formulated and evaluated. The results showed that the addition of silicificated microcrystalline cellulose to chitosan improves the compressibility by increasing the energy of plastic deformation responsible for the formation of bonds and their strength. Furthermore, the sensitivity to the lubricant is markedly reduced due to the competitive inhibition of binding sites by colloidal silica contained in the silicificated microcrystalline cellulose.

The consistency and shearing behaviour of a gel layer formed after exposing the chitosan-based matrix tablets to PBS pH 6.8 can be mediated by the type and concentration of the retarding components used. Both sodium alginate and hypromellose support the adhesion of chitosan-based matrix tablets to the intestinal mucosa. The addition of silicified microcrystalline cellulose does not have any negative impact on mucoadhesion. 

The formulated chitosan-based matrix systems with silicificated microcrystalline cellulose represent a suitable basis for incorporation of an active ingredient whose delayed release can be achieved by the functionality of the chitosan–sodium alginate complex.

## Figures and Tables

**Figure 1 polymers-13-03636-f001:**
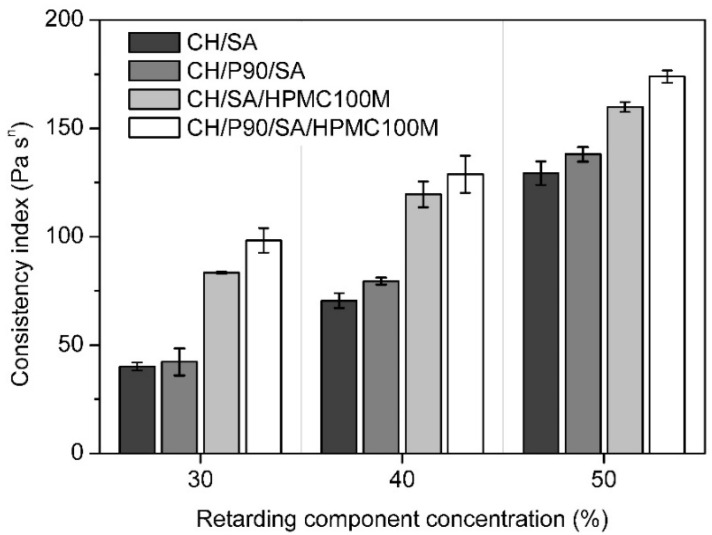
Consistency of the dispersions of tableting materials in PBS pH 6.8. CH—chitosan; P90—silicificated microcrystalline cellulose Prosolv SMCC 90; SA—sodium alginate; HPMC100M—hypromellose Methocel K100M. Data are expressed as average values ± SD of three tests.

**Figure 2 polymers-13-03636-f002:**
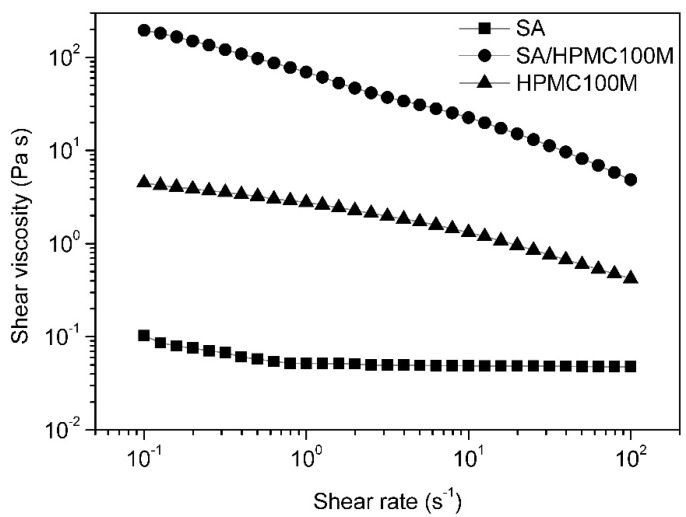
Viscosity curves of 2% dispersions of retarding excipients in PBS pH 6.8. SA—sodium alginate; HPMC100Mhypromellose Methocel K100M; SA/HPMC100M—mixture of sodium alginate and hypromellose in a ratio of 1:1.

**Figure 3 polymers-13-03636-f003:**
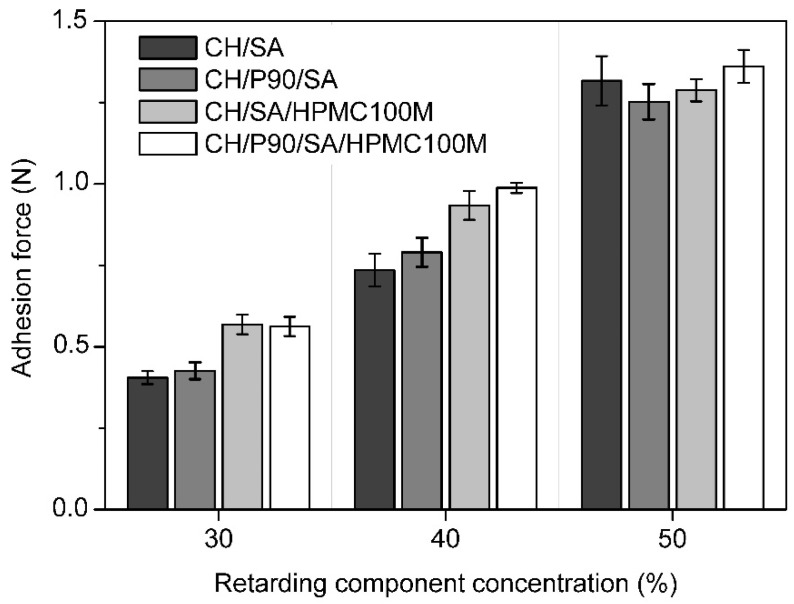
Mucoadhesive properties of the chitosan—based matrix tablets treated with PBS pH 6.8. CH—chitosan; P90—Prosolv SMCC 90; SA—sodium alginate; HPMC100M—Methocel K100M. Data are expressed as average values ± SD of five tests.

**Table 1 polymers-13-03636-t001:** Composition of tableting materials with chitosan.

Tableting Material	CH (%)	SA (%)	HPMC100M (%)	Mgst (%)
F1	100	-	-	-
F2	70	30	-	-
F3	60	40	-	-
F4	50	50	-	-
F1L	99	-	-	1
F2L	69	30	-	1
F3L	59	40	-	1
F4L	49	50	-	1
F5	70	15	15	-
F6	60	20	20	-
F7	50	25	25	-
F5L	69	15	15	1
F6L	59	20	20	1
F7L	49	25	25	1

CH—chitosan; SA—sodium alginate; Mgst—magnesium stearate; HPMC100M—hypromellose Methocel K100M.

**Table 2 polymers-13-03636-t002:** Composition of tableting materials with chitosan and P90 at a ratio of 3:1.

Tableting Material	CH + P90 3:1 (%)	SA (%)	HPMC100M (%)	Mgst (%)
FP1	100	-	-	-
FP2	70	30	-	-
FP3	60	40	-	-
FP4	50	50	-	-
FP1L	99	-	-	1
FP2L	69	30	-	1
FP3L	59	40	-	1
FP4L	49	50	-	1
FP5	70	15	15	-
FP6	60	20	20	-
FP7	50	25	25	-
FP5L	69	15	15	1
FP6L	59	20	20	1
FP7L	49	25	25	1

CH—chitosan; P90—silicificated microcrystalline cellulose Prosolv SMCC 90; SA—sodium alginate; HPMC100M—hypromellose Methocel K100M; Mgst—magnesium stearate.

**Table 3 polymers-13-03636-t003:** Values of energy profile of compression and plasticity—tableting materials with chitosan.

Tableting Material	Emax ± SD (J)	E1 ± SD (J)	E2 ± SD (J)	E3 ± SD (J)	Pl ± SD (J)
F1	6.31 ± 0.44	4.07 ± 0.44	1.76 ± 0.05	0.48 ± 0.01	78.66 ± 0.52
F2	4.50 ± 0.37	2.34 ± 0.26	1.38 ± 0.12	0.47 ± 0.01	78.05 ± 0.91
F3	3.84 ± 0.39	1.84 ± 0.20	1.54 ± 0.18	0.46 ± 0.01	77.03 ± 1.78
F4	4.07 ± 0.37	1.90 ± 0.20	1.71 ± 0.17	0.46 ± 0.01	78.68 ± 1.25
F1L	5.21 ± 0.25	3.09 ± 0.16	1.62 ± 0.08	0.49 ± 0.02	76.63 ± 0.62
F2L	4.05 ± 0.14	2.11 ± 0.10	1.47 ± 0.05	0.47 ± 0.00	75.82 ± 0.54
F3L	3.60 ± 0.17	1.73 ± 0.14	1.41 ± 0.04	0.46 ± 0.00	75.42 ± 0.53
F4L	3.43 ± 0.23	1.58 ± 0.18	1.39 ± 0.05	0.46 ± 0.01	75.31 ± 0.52
F5	5.77 ± 0.32	3.63 ± 0.30	1.68 ± 0.05	0.45 ± 0.01	78.73 ± 0.45
F6	5.72 ± 0.44	3.63 ± 0.43	1.63 ± 0.02	0.46 ± 0.00	78.10 ± 0.28
F7	5.60 ± 0.33	3.53 ± 0.31	1.62 ± 0.03	0.45 ± 0.01	78.13 ± 0.28
F5L	5.88 ± 0.17	3.87 ± 0.14	1.55 ± 0.05	0.46 ± 0.01	77.24 ± 0.32
F6L	5.73 ± 0.19	3.66 ± 0.17	1.60 ± 0.05	0.46 ± 0.01	77.54 ± 0.36
F7L	4.99 ± 0.31	3.03 ± 0.30	1.50 ± 0.03	0.46 ± 0.01	76.76 ± 0.25

Emax—total energy of compression; E1—energy of precompression; E2—energy of plastic deformation; E3—energy of elastic deformation; Pl—plasticity. Data are expressed as average values ± SD of 10 tests.

**Table 4 polymers-13-03636-t004:** Values of energy profile of compression and plasticity—tableting materials with chitosan and P90 3:1.

Tableting Material	Emax ± SD (J)	E1 ± SD (J)	E2 ± SD (J)	E3 ± SD (J)	Pl ± SD (J)
FP1	7.44 ± 0.13	4.98 ± 0.11	1.99 ± 0.04	0.47 ± 0.01	80.88 ± 0.22
FP2	5.78 ± 0.14	3.49 ± 0.13	1.78 ± 0.01	0.45 ± 0.00	79.69 ± 0.18
FP3	5.46 ± 0.08	3.28 ± 0.08	1.73 ± 0.01	0.45 ± 0.00	79.38 ± 0.14
FP4	5.04 ± 0.12	2.91 ± 0.11	1.68 ± 0.01	0.45 ± 0.01	79.00 ± 0.15
FP1L	6.17 ± 0.05	3.84 ± 0.05	1.91 ± 0.02	0.43 ± 0.00	81.70 ± 0.13
FP2L	5.43 ± 0.14	3.31 ± 0.13	1.66 ± 0.01	0.46 ± 0.00	78.18 ± 0.15
FP3L	4.97 ± 0.09	2.90 ± 0.08	1.62 ± 0.01	0.46 ± 0.00	77.99 ± 0.14
FP4L	4.46 ± 0.09	2.46 ± 0.08	1.55 ± 0.01	0.46 ± 0.02	77.22 ± 0.24
FP5	6.16 ± 0.22	3.89 ± 0.23	1.83 ± 0.02	0.45 ± 0.01	80.40 ± 0.25
FP6	5.98 ± 0.30	3.80 ± 0.31	1.75 ± 0.03	0.43 ± 0.01	80.26 ± 0.43
FP7	6.09 ± 0.20	3.92 ± 0.20	1.74 ± 0.02	0.43 ± 0.01	80.36 ± 0.21
FP5L	6.25 ± 0.18	4.07 ± 0.18	1.74 ± 0.02	0.44 ± 0.00	79.84 ± 0.25
FP6L	6.16 ± 0.12	4.03 ± 0.11	1.68 ± 0.01	0.44 ± 0.00	79.15 ± 0.22
FP7L	5.93 ± 0.18	3.86 ± 0.17	1.64 ± 0.03	0.44 ± 0.00	78.94 ± 0.30

Emax—total energy of compression; E1—energy of precompression; E2—energy of plastic deformation; E3—energy of elastic deformation; Pl—plasticity. Data are expressed as average values ± SD of 10 tests.

**Table 5 polymers-13-03636-t005:** Values of tensile strength and LSR of tableting materials.

Tableting Material	TS ± SD (MPa)	LSR ± SD	Tableting Material	TS ± SD (MPa)	LSR ± SD
F1	1.212 ± 0.072	0.40 ± 0.02	FP1	1.909 ± 0.160	0.01 ± 0.03
F2	0.682 ± 0.035	0.30 ± 0.02	FP2	1.312 ± 0.068	0.26 ± 0.02
F3	0.554 ± 0.054	0.27 ± 0.03	FP3	1.302 ± 0.079	0.44 ± 0.01
F4	0.484 ± 0.053	0.29 ± 0.05	FP4	0.999 ± 0.077	0.53 ± 0.01
F1L	0.730 ± 0.047	-	FP1L	1.885 ± 0.036	-
F2L	0.476 ± 0.037	-	FP2L	0.973 ± 0.051	-
F3L	0.407 ± 0.020	-	FP3L	0.732 ± 0.029	-
F4L	0.341 ± 0.059	-	FP4L	0.471 ± 0.024	-
F5	1.873 ± 0.117	0.16 ± 0.02	FP8	2.756 ± 0.118	0.10 ± 0.01
F6	1.977 ± 0.141	0.24 ± 0.02	FP9	2.631 ± 0.161	0.15 ± 0.03
F7	2.016 ± 0.061	0.30 ± 0.01	FP10	2.457 ± 0.054	0.17 ± 0.02
F5L	1.571 ± 0.062	-	FP8L	2.494 ± 0.030	-
F6L	1.510 ± 0.064	-	FP9L	2.240 ± 0.159	-
F7L	1.404 ± 0.057	-	FP10L	2.030 ± 0.124	-

TS—tensile strength of tablets; LSR—lubricant sensitivity ratio. Data are expressed as average values ± SD of 10 tests.

**Table 6 polymers-13-03636-t006:** Values of power law index n (−) of the dispersions of tableting materials in PBS pH 6.8.

Retarding Component	n Values for CH	n Values for CH/P90
SA 30%	0.5847 ± 0.005	0.5356 ± 0.047
SA 40%	0.5608 ± 0.012	0.5305 ± 0.011
SA 50%	0.5050 ± 0.007	0.4937 ± 0.006
SA/HPMC100M 30%	0.4665 ± 0.014	0.4950 ± 0.011
SA/HPMC100M 40%	0.4702 ± 0.019	0.4867 ± 0.018
SA/HPMC100M 50%	0.4642 ± 0.002	0.4389 ± 0.002

CH—chitosan; SA—sodium alginate; CH/P90—mixture of chitosan and Prosolv SMCC 90 in a ratio of 3:1; SA/HPMC100M—mixture of sodium alginate and Methocel K100M in a ratio of 1:1; n—power law index being from 0 for highly shear-thinning materials to 1 for Newtonian materials. Data are expressed as average values ± SD of three tests.

## Data Availability

The data presented in this study are available on request from the corresponding author.

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
