# Peer review of "A Study of Compressibility, Compactability and Mucoadhesivity of Tableting Materials for Matrix Systems Based on Chitosan"

_polymers, 2021, doi:10.3390/polym13213636_

Round 1
Reviewer 1 Report
It is a very good study with overall adequate presentation of experimental results. Some additions are needed:
1) Authors should further emphasize on the novelty of their work.
2) Some minor typos, grammar and syntax errors should be carefully revised and corrected accordingly.
3) Reference can be even more updated (more recent relative works).
Reviewer 2 Report
The present work highlights the chitosan and derivatives compressed tablets for drug delivery in the tract intestine. The study targets an exciting and needed topic around the world. However, several things need to be attended to before the publication. All the suggestions are highlighted in the manuscript attached. Essential suggestions are to review the grammar and English style used, add more discussion to the results and discussion section, add statistical analysis and description of the methodology used for that analysis.

Reviewer 3 Report
Dear authors!
The manuscript present new information that will open potential for chitosan application in oral drug administration. Some minor corrections required:
- In introduction section the more application of chitosan should be demonstrated. Please, use the recent research and review papers - 1557/s43578-021-00358-4, 10.3390/biomedicines9060588, 10.3390/ijms21020487
- In materials section the statistics methods description required
- The chapter before the subsection 3.1. is not need. It is just repeat the information from the introduction and materials section.
- In results description the significance of difference should be provided
Round 2
Reviewer 2 Report
I appreciate that almost all the suggestions were addressed. I want to emphasize that authors should highlight the novelty of the work more in the introduction section instead of the conclusions section. However, the manuscript is ready for publication.